# Human albumin enhances the pathogenic potential of *Candida glabrata* on vaginal epithelial cells

**Marina Pekmezovic**[1], **Ann-Kristin Kaune**[1¤], **Sophie Austermeier**[1], **Sophia U. J. Hitzler**[2], **Selene Mogavero**[1], **Hrant Hovhannisyan**[3,4], **Toni Gabaldón**[3,4,5], **Mark S. Gresnigt**[2]\*, **Bernhard Hube**[1,6]\*

**1** Department of Microbial Pathogenicity Mechanisms, Leibniz Institute for Natural Product Research and Infection Biology, Hans Knoell Institute, Jena, Germany, **2** Junior Research Group Adaptive Pathogenicity Strategies, Leibniz Institute for Natural Product Research and Infection Biology, Hans Knoell Institute, Jena, Germany, **3** Life Sciences Department, Barcelona Supercomputing Center (BSC), Barcelona, Spain, **4** Mechanisms of Disease Department, Institute for Research in Biomedicine (IRB), Barcelona, Spain, **5** Institució Catalana de Recerca i Estudis Avançats (ICREA), Barcelona, Spain, **6** Institute of Microbiology, Friedrich Schiller University, Jena, Germany

¤ Current Address: University of Aberdeen, Institute of Medical Sciences, Foresterhill, Aberdeen, United Kingdom

\* mark.gresnigt@leibniz-hki.de (MSG); bernhard.hube@leibniz-hki.de (BH)

**Data Availability Statement:** All codes, packages and their versions are available at https://github.com/Gabaldonlab/C_glabrata_with_albumin. All transcriptome data analysis results, including figures and supplementary materials are fully

## Abstract

The opportunistic pathogen *Candida glabrata* is the second most frequent causative agent of vulvovaginal candidiasis (VVC), a disease that affects 70–75% of women at least once during their life. However, *C. glabrata* is almost avirulent in mice and normally incapable of inflicting damage to vaginal epithelial cells *in vitro*. We thus proposed that host factors present *in vivo* may influence *C. glabrata* pathogenicity. We, therefore, analyzed the impact of albumin, one of the most abundant proteins of the vaginal fluid. The presence of human, but not murine, albumin dramatically increased the potential of *C. glabrata* to damage vaginal epithelial cells. This effect depended on macropinocytosis-mediated epithelial uptake of albumin and subsequent proteolytic processing. The enhanced pathogenicity of *C. glabrata* can be explained by a combination of beneficial effects for the fungus, which includes an increased access to iron, accelerated growth, and increased adhesion. Screening of *C. glabrata* deletion mutants revealed that Hap5, a key regulator of iron homeostasis, is essential for the albumin-augmented damage potential. The albumin-augmented pathogenicity was reversed by the addition of iron chelators and a similar increase in pathogenicity was shown by increasing the iron availability, confirming a key role of iron. Accelerated growth not only led to higher cell numbers, but also to increased fungal metabolic activity and oxidative stress resistance. Finally, the albumin-driven enhanced damage potential was associated with the expression of distinct *C. glabrata* virulence genes. Transcriptional responses of the epithelial cells suggested an unfolded protein response (UPR) and ER-stress responses combined with glucose starvation induced by fast growing *C. glabrata* cells as potential mechanisms by which cytotoxicity is mediated.Collectively, we demonstrate that albumin augments the pathogenic potential of

reproducible using the scripts available at https://github.com/Gabaldonlab/C_glabrata_with_albumin. Raw sequencing data is submitted under project accession number PRJNA745548 to SRA database.

**Funding:** M.P., H.H., T.G., and B.H. received funding from the European Union Horizon 2020 research and innovation program under the Marie Sklodowska-Curie grant agreement No 642095 (OPATHY). A.K. and B. H. received support from the European Union Horizon 2020 research and innovation program under the Marie Sklodowska-Curie grant agreement No 812969 (FunHoMic). S. A. and B.H. were supported by funding from the European Union's Horizon 2020 research and innovation program under grant agreement No 847507 (HDM-FUN). B.H. is further supported by the DFG within the Collaborative Research Centre (CRC)/Transregio (TRR) 124 "FungiNet" project C1 (DFG project number 210879364) and the Balance of the Microverse Cluster (Germany´s Excellence Strategy – EXC 2051 – Project-ID 390713860). M. S.G. was supported by the German Research Foundation (Deutsche Forschungsgemeinschaft - DFG) Emmy Noether Program (project no. 434385622 / GR 5617/1-1), and a Research Grant 2019 from the European Society of Clinical Microbiology and Infectious Diseases (ESCMID). The funders had no role in study design, data collection and analysis, decision to publish, or preparation of the manuscript. M.P. and H.H. received salary from grant agreement No 642095 (OPATHY) (2016-2019). A.K. received salary from grant agreement No 812969 (FunHoMic) (2019-2022).

**Competing interests:** The authors have declared that no competing interests exist.

*C. glabrata* during interaction with vaginal epithelial cells. This suggests a role for albumin as a key player in the pathogenesis of VVC.

## Author summary

*Candida glabrata* is the overall second causative species of candidiasis in humans, but little is known about the pathogenicity mechanisms of this yeast. *C. glabrata* is capable of causing lethal systemic candidiasis mostly in elderly immunocompromised patients, but is also a frequent cause of vulvovaginal candidiasis. These clinical insights suggest that *C. glabrata* has a high virulence potential, yet little pathogenicity is observed in both *in vitro* and *in vivo* infection models. The finding that human albumin, the most abundant protein in the human body, is boosting *C. glabrata* pathogenicity *in vitro* provides novel insights into *C. glabrata* pathogenicity mechanisms and shows that the presence of distinct human factors can have a significant influence on the virulence potential of a pathogenic microbe.

## Introduction

Vulvovaginal candidiasis (VVC) is the second most common vaginal infection, occurring in 70–75% of women at least once during their lifetime [1–3]. *Candida albicans* is the major cause of VVC, but the incidence of infections caused by non-*albicans Candida* species is increasing [4]. Even though *C. glabrata* is found in many studies as the most common non-*albicans* species causing VVC [4–7], its pathogenicity mechanisms are unresolved. Despite being able to cause both mucosal and systemic infections in clinical settings, *C. glabrata* is often avirulent in mice [8] and causes almost no epithelial damage *in vitro* [9]. We thus proposed that these models may lack specific host factors, which are crucial for triggering the full pathogenic potential of *C. glabrata*.

Albumin is the most abundant serum protein in humans, and is present in many body fluids, such as sweat, tears, saliva, and the vaginal fluid [10–13]. Therefore, commensal and pathogenic microorganisms, including *C. glabrata*, constantly interact with albumin while colonizing and infecting the vaginal mucosa. Albumin carries out diverse, well-characterized, functions in the human body [11,14,15] and several new functions have been discovered, especially in the context of host-pathogen interactions [16–20]. For example, specific albumin properties, such as binding and transport of several molecules, can be exploited by pathogenic microorganisms [16,21], yet such strategies are poorly explored in the context of mucosal infections. We hypothesized that albumin could serve as a nutrient source, as a bridging protein to enhance adhesion to and invasion into epithelial cells, or to help in resisting host defenses. For example, it may protect against antimicrobial effectors or it may mediate immune evasion.

In this study, we investigated whether and how albumin plays a role during *C. glabrata* infection of human vaginal epithelium, using an *in vitro* VVC model consisting of vaginal epithelial cells [9]. We discovered that *C. glabrata* can cause epithelial damage in the presence of human albumin and dissected the mechanisms leading to its increased pathogenicity. Our observations suggest that the presence of a distinct human factor can have a significant impact on the virulence potential of *C. glabrata*.

## Results

### *Candida glabrata* is causing high epithelial damage in the presence of human albumin

During *in vitro* infection of vaginal epithelial cells, *C. glabrata* causes little to no damage to host cells (Fig 1a; [9]). However, the presence of human albumin increased the damage potential of *C. glabrata* to a level comparable to *C. albicans*, previously reported as the most highly damaging *Candida* species (Fig 1a; [9]). Bovine, but not murine, albumin exhibited the same effect as human albumin on *C. glabrata* damage potential (Fig 1b). These results were confirmed by testing human and murine albumin obtained from different manufacturers, as well as various *C. glabrata* clinical isolates (S1 Fig). These differences could potentially be related to a higher similarity between bovine and human albumin protein sequences, compared to murine albumin (S1 Fig and S1 File).

To dissect the effect of albumin on the interaction of *C. glabrata* with epithelial cells, we quantified fungal attributes potentially associated with host cell cytotoxicity. Apart from damage, *C. glabrata* exhibited also increased adhesion to vaginal epithelial cells (Fig 1c) and showed increased proliferation on epithelial cells in the presence of human, but not murine, albumin (Fig 1d).

The increased adhesion can be explained by the increased expression of adhesins in the presence of albumin. Transcriptional profiling during infection of vaginal epithelial cells revealed that *C. glabrata* up-regulated the adhesin genes *AWP2* and *AWP6* [22,23], as well as several further putative adhesin genes [24], at the early infection stage in the presence of albumin as compared to infection without it (S1A Fig).

The increased damage could be a consequence of the increased numbers of *C. glabrata* cells infecting epithelial cells. Indeed, infections with different inocula (multiplicity of infection (MOI) of 1, 5, 10, 50 and 100) revealed that damage increased even in albumin-free conditions (Fig 1e). However, the damage induced by *C. glabrata* (MOI 1) in albumin-containing medium corresponded to damage by a MOI of 50 to MOI of 100 in albumin-free medium, even though the *C. glabrata* number only increased approximately three-fold (Fig 1d). These data suggest that albumin increases *C. glabrata* pathogenicity, which can only partially be explained by increased proliferation. We further demonstrated, that the increased damage in the presence of albumin requires direct physical contact between *C. glabrata* and host cells (Fig 1f). Higher MOIs in transwell inserts did not induce damage (Fig 1f), which excludes the possibility that nutrient consumption alone causes the increased damage.

### Host cell uptake and proteolytic processing of albumin

Since albumin augmented *C. glabrata* proliferation during epithelial infection, we hypothesized that albumin could serve as an additional nutrient source that helps to sustain enhanced growth rates. However, albumin did not increase *C. glabrata* growth in the absence of host cells (Fig 2a), suggesting that activities of the epithelial cells are essential for the increased fungal growth. The increased growth was independent of the direct contact to the host cells, since *C. glabrata* in transwell inserts still showed increased growth when epithelial cells were present (Fig 2b). Although *C. glabrata* expresses GPI-anchored (Yps) proteases on the cellular surface [25], it does not possess proteases that are secreted into the extracellular space to degrade albumin [26]. It may therefore rely on host proteolytic activity to use albumin as a nutrient source [21]. Despite the presence of albumin, *C. glabrata* failed to cause epithelial damage when proteases were inhibited using a protease inhibitor cocktail (Fig 2c). In addition, protease inhibitors prevented the albumin-mediated increase in proliferation rates (Fig 2d). Previous studies have shown that albumin is taken up by host cells through endocytosis and transported to

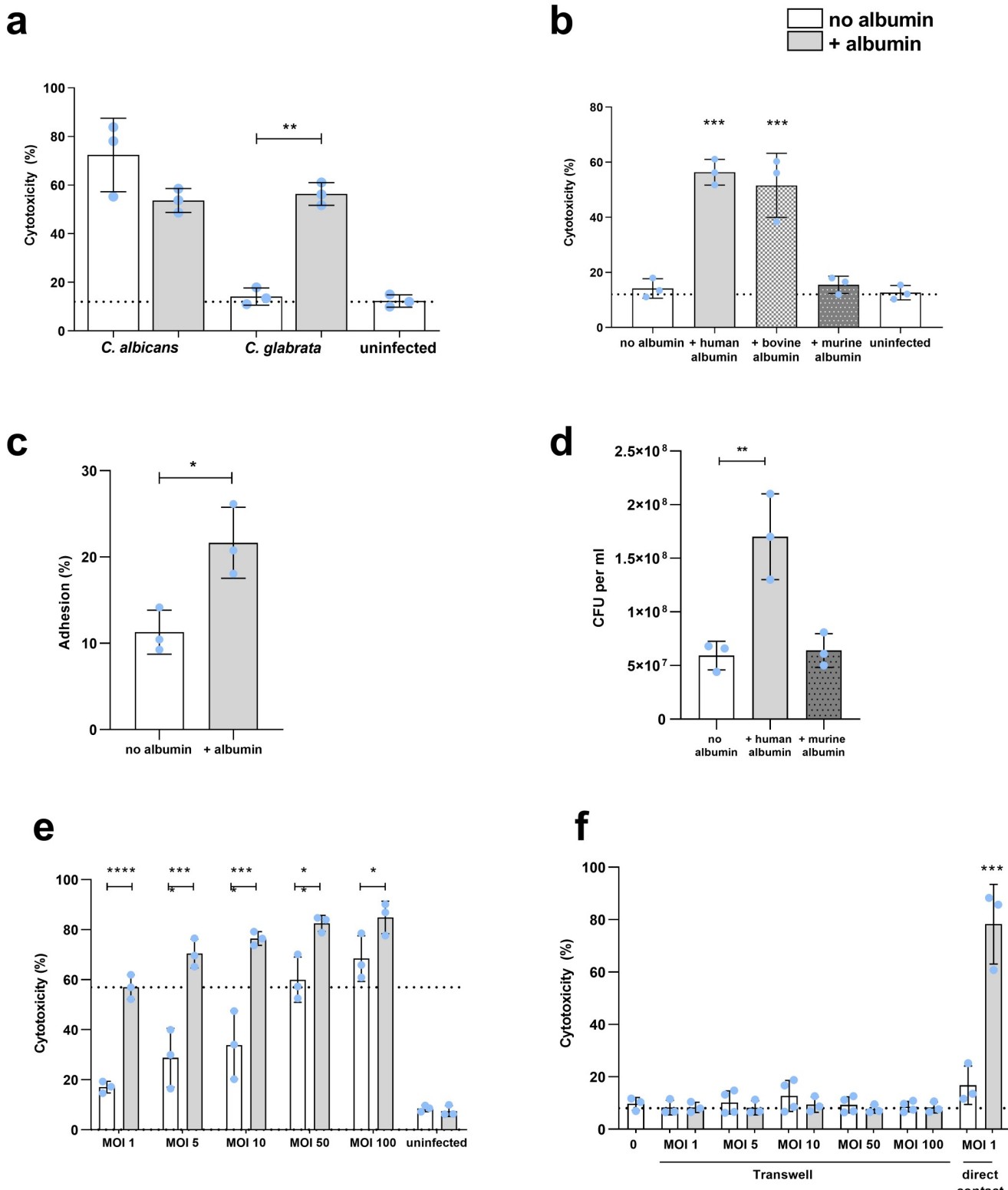

**Fig 1. Human, but not murine, albumin increases growth and epithelial damage caused by *C. glabrata*. (a)** Damage of A-431 cells infected with *C. albicans* and *C. glabrata* with or without albumin. **(b)** Damage of A-431 cells infected with *C. glabrata* with or without human, bovine, or murine albumin. **(c)** Adhesion of *C. glabrata* to A-431 cells with or without albumin. The percentage of adhered *C. glabrata* was calculated relative to the inoculum used for infection. **(d)** *C. glabrata* growth 24 h post infection of A-431 cells with or without human or murine albumin. **(e-f)** Damage of A-431 cells infected **(e)** in

direct contact or (**f**) in transwells preventing direct contact with different multiplicities of infection (MOI) of *C. glabrata* with or without albumin. Infection with MOI 1 in direct contact was used as a control. All values are presented as mean ± SD. Damage was recorded by measuring the lactate dehydrogenase activity in the supernatant and presented as percentage of a full lysis control (A-431 treated with Triton X-100). The dotted line represents damage from uninfected A-431 cells (a, b, f) or damage upon MOI 1 in the presence of albumin (e). Albumin was always used at a 5 mg/mL concentration. One-way ANOVA (a-d) or two-way ANOVA (e-f) were used to calculate statistically significant differences. Statistical significance is indicated as: *, $p \leq 0.05$; **, $p \leq 0.01$; ***, $p \leq 0.001$; ****, $p \leq 0.0001$.

lysosomal compartments, where it is degraded [27]. Using human albumin conjugated with Alexa Fluor 647, we demonstrated that albumin localizes in intracellular vesicles inside the vaginal epithelial cells (Fig 2e). Inhibition of phagocytosis-like and macropinocytosis processes prevented the increased damage by *C. glabrata* in the presence of albumin, indicating that albumin uptake by host cells is required for the increased damage potential of *C. glabrata* (Fig 2f).

A number of albumin-binding proteins or receptors responsible for albumin uptake have been identified and characterized in various tissues and cell lines (reviewed in [11]). Transcriptional profiling of vaginal epithelial cells revealed a specific regulation of albumin-binding proteins and receptors. For example, genes encoding the albumin receptors cubilin (*CUBN*) and the neonatal Fc receptor (*FCRN*) were up-regulated in the presence of albumin, independently of infection, while *SPARC,* coding for an albumin binding protein, was up-regulated only during infection at 24 h, but not in uninfected control conditions (24 c) (Fig 2g).

## Metabolic adaptation and oxidative stress as a transcriptional core response of *C. glabrata*

To elucidate how the pathogenic potential of *C. glabrata* is modified in the presence of albumin, we performed transcriptional profiling of *C. glabrata* at different time points during infection. Principal component analysis (PCA) revealed that samples clustered by replicates of each time point, indicating reliable reproducibility and dynamic change of the transcriptional response over time (S2B Fig). Gene-Ontology (GO) term enrichment analysis revealed a modulation of the fungal metabolism during the course of infection. Various fungal genes associated with catabolic processes of different sugars and fatty acids were up-regulated (Fig 3). Strikingly, oxidative stress responses were activated throughout the infection (Fig 3). This is further supported by the upregulation of the genes involved in trehalose biosynthesis, previously shown to be protective during severe oxidative stress in *C. albicans* [28].

## Infection in the presence of albumin induces oxidative stress resistance in *C. glabrata*

As the genes involved in response to oxidative stress were significantly up-regulated (Fig 3), we checked whether ROS production by epithelial cells was increased during infection in the presence of albumin and therefore causing an additional stress for the fungus. Interestingly, even lower ROS levels were detected when albumin was present (Fig 4a). Moreover, there was no difference in ROS production of infected and uninfected epithelial cells, independently of albumin (Fig 4a). This observation suggests that the up-regulation of oxidative stress resistance genes in *C. glabrata* is not triggered by host-derived ROS. We thus proposed that the up-regulation of oxidative stress resistance genes may be an intrinsic mechanism to cope with increased ROS production within the yeast cells associated with increased metabolic rates. Consequently, *C. glabrata* should show an increased resistance against oxidative stress. Indeed, *C. glabrata* cells co-incubated with epithelial cells in the presence of human albumin for 3 hours showed significantly increased resistance to subsequent $H_2O_2$-induced oxidative stress, compared to *C. glabrata* cells incubated with epithelial cells in medium without albumin (Fig 4b).

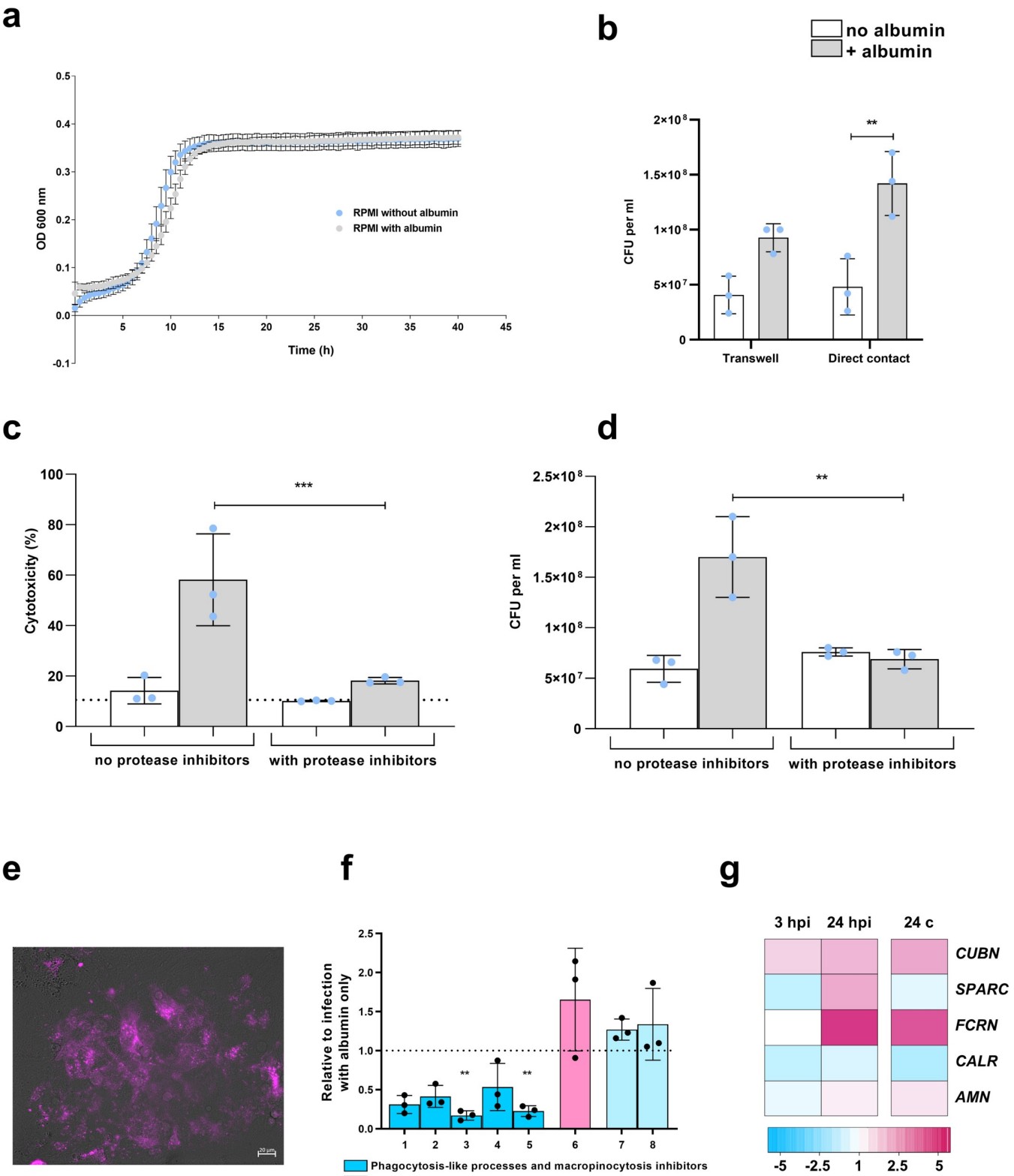

**Fig 2. Role of host cell presence and activity in albumin-augmented growth of *C. glabrata*.** (**a**) Growth of *C. glabrata* with or without albumin, in the absence of host cells. Optical density was measured every 30 min for 40 h at 600 nm. (**b**) *C. glabrata* growth 24 h post infection of A-431 cells in transwell system and in direct contact between *C. glabrata* and A-431 cells. (**c**) Damage of A-431 cells infected with *C. glabrata* with or without the addition of protease

inhibitor cocktail with or without albumin. The dotted line represents damage from uninfected A-431 cells. **(d)** *C. glabrata* growth 24 h post infection of A-431 cells with or without the addition of protease inhibitor cocktail with or without albumin. **(e)** Albumin localization in infected A-431 cells after 6 h, tracked using human albumin-Alexa Fluor 647 (1 mg/mL). Scale bar corresponds to 20 μm. **(f)** Damage of A-431 cells infected with *C. glabrata* with albumin in the presence of different endocytosis inhibitors. Endocytosis inhibitors used: amiloride 0.5 mM (**1**), colchicine 50 μM (**2**), nocodazole 66 μM (**3**), cytochalasin D 0.5 (**4**) and 10 μM (**5**), genistein 125 μM (**6**), methy-β-cyclodextrin 3 mM (**7**), and simvastatin 10 μM (**8**). Values are presented as relative to infection in albumin-containing medium without any inhibitor (dotted line). **(g)** Expression of selected human genes involved in albumin binding and uptake at 3 and 24 hpi and 24 h control (24 c) in the presence of albumin, derived from RNA-Seq data (see Material and Methods). Presented values are expressed as $\log_2$ fold changes of expression compared to host cells in medium only at the time point 0. All values are presented as mean ± SD. Damage was recorded by measuring the lactate dehydrogenase activity in the supernatant and presented as percentage of a full lysis control (A-431 treated with Triton X-100). Albumin was always used at a 5 mg/mL concentration. One-way ANOVA was used to calculate statistically significant differences. Statistical significance is indicated as: $^{**}$, $p \leq 0.01$; $^{***}$, $p \leq 0.001$.

## Iron regulation mediates albumin-augmented damage by *C. glabrata*

To identify critical *C. glabrata* genes that regulate or mediate the increased epithelial damage in the presence of albumin, a *C. glabrata* deletion mutant library [29] was screened. Out of 623

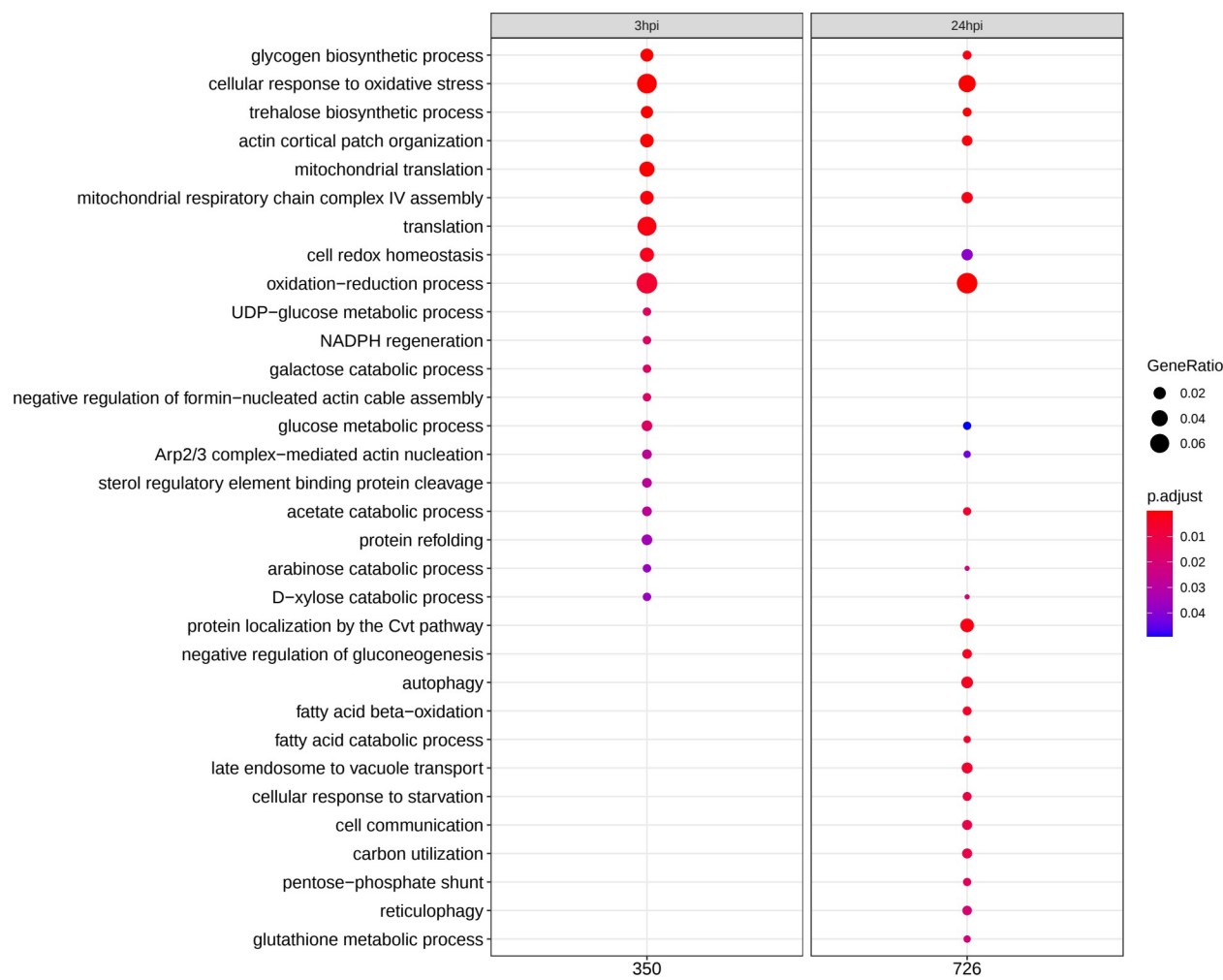

**Fig 3. Transcriptome dynamics of *C. glabrata* upon *in vitro* infection of vaginal epithelial cells in the presence of albumin.** Gene-Ontology (GO) term enrichment analysis for up-regulated genes (category "Biological Process") of *C. glabrata* at 3 and 24 hours post-infection (hpi). Numbers underneath the plot correspond to "counts" of clusterProfiler Bioconductor package, i.e. the total number of genes associated with GO categories. GeneRatio corresponds to the ratio between the number of genes enriched in a given category and "counts". Only significant ($p<0.05$) GO enrichments are shown. Adjustment of $p$-values is done by the Benjamini-Hochberg procedure.

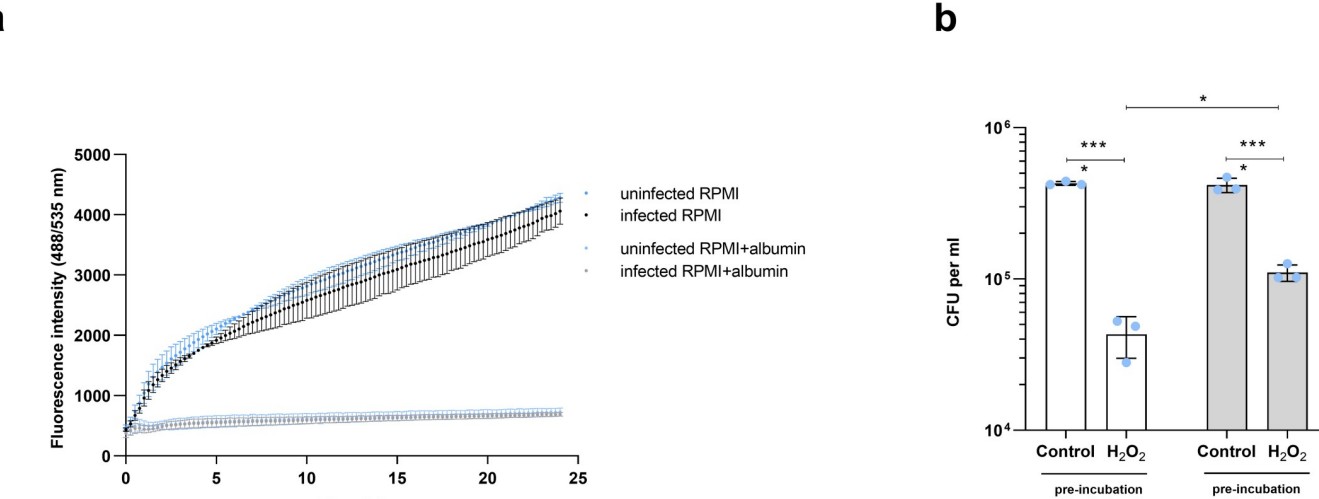

**Fig 4. Oxidative stress during *C. glabrata* infection in the presence of albumin. (a)** Reactive oxygen species (ROS) activity in A-431 cells infected with *C. glabrata* with or without albumin. Fluorescence intensity (485/535 nm) was measured over 24 h using ROS indicator 2',7'–dichlorofluorescein diacetate (DCFDA). **(b)** Growth of *C. glabrata* upon the exposure to 10 mM $H_2O_2$ or medium only (control), after 3-hour pre-incubation with A-431 cells in the presence or absence of albumin. All values are presented as mean ± SD. One-way ANOVA was used to calculate statistically significant differences. Albumin was always used at a 5 mg/mL concentration. Statistical significance is indicated as: ****, $p \leq 0.0001$.

mutants tested, albumin did not augment pathogenic potential in 37 mutant strains (Fig 5a). Most of those mutants (33/37) are known to have a reduced fitness from previous phenotyping studies and were excluded from further analyses [29]. The four remaining mutants (*cdc10Δ*, *hap5Δ*, *ost3Δ* and *ost6Δ*) were further analyzed. Their growth in the absence of albumin was tested to exclude a growth deficiency as the cause of the reduced damage capacity (Fig 5b). Only *hap5Δ* demonstrated a similar growth capacity as the WT (Fig 5b), but lacked the increased damage potential in the presence of albumin (Fig 5c). Comparison of the damage capacities of the parental non-auxotrophic ATCC 2001 WT strain and the *hap5Δ* mutant in a non-auxotrophic background with the auxotrophic *hap5Δ* mutant (c_ *hap5Δ*) and isogenic WT from the collection (c_WT) revealed that the increased damage phenotype by albumin is independent of auxothrophies (Fig 5c).

Hap5 is a regulator of iron homeostasis [30–32], which suggests a connection between albumin-augmented damage and iron homeostasis or availability. Interestingly, epithelial damage induced by *C. glabrata* in DMEM-F12 medium, which contains additional nutrients such as ferric nitrate and ferric sulfate, was significantly higher compared to infections in the presence of RPMI medium (used in all assays presented above) even in the absence of albumin (Fig 6a). Albumin can bind heme and was also demonstrated to deliver iron to *C. albicans* via the CFEM hemophore relay network [21,33]. Although the CFEM hemophore system is not described in *C. glabrata*, albumin may deliver iron via alternative unknown mechanisms, leading to increased growth and damage potential of *C. glabrata* in the presence of albumin. In blood, saliva, and vaginal fluid, transferrin sequesters any free iron due to its high affinity. Unlike other fungal pathogens, such as *C. albicans*, *C. glabrata* cannot exploit transferrin as an iron source [32]. In the presence of transferrin, the albumin-augmented *C. glabrata* growth and damage potential was neutralized (Fig 6b and 6c). Similarly, neither albumin-augmented damage nor growth was observed when iron was sequestered during infection with the iron chelator BPS (Fig 6b and 6c). To test whether an increased iron availability can underlie the augmented epithelial damage by *C. glabrata*, we supplemented different concentrations of

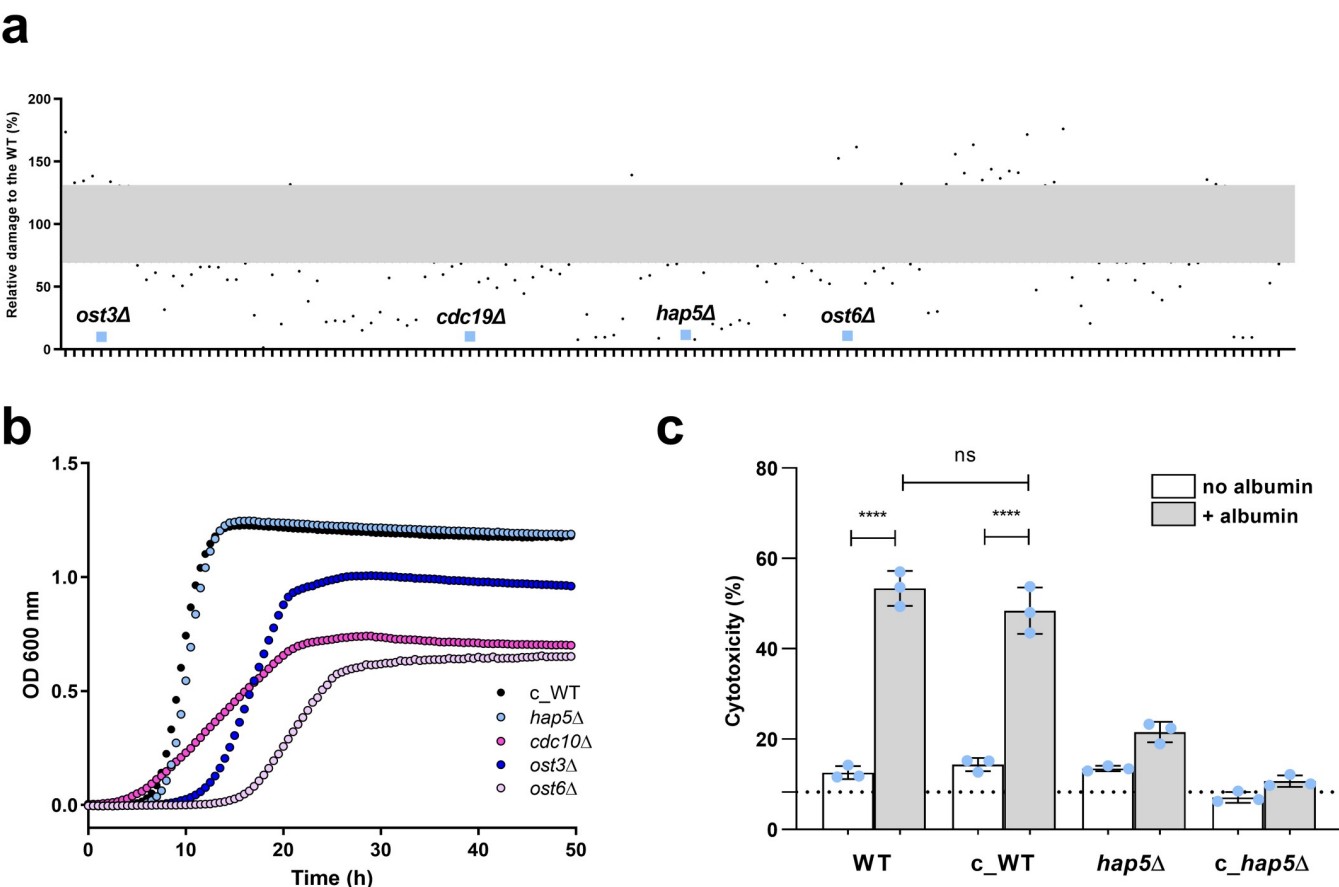

**Fig 5. *C. glabrata* deletion mutant library screening. (a)** Damage of A-431 cells infected with *C. glabrata* deletion mutants. Values are presented as relative to the collection parental wild type (c_WT) damage level. Each dot corresponds to a different mutant. Only mutants exhibiting damage lower that 70% of the c_WT or higher than 130% are shown. The complete mutant screening dataset is available in S1 Table. **(b)** Growth of selected *C. glabrata* mutant strains in YPD compared to the WT. Optical density was measured every 30 min for 50 h at 600 nm. **(c)** Damage of A-431 cells infected with WT, c_WT, *hap5*Δ mutant in WT background and *hap5*Δ mutant in c_WT background (c_ *hap5*Δ). All values are presented as mean ± SD. Damage was recorded by measuring LDH in the supernatant and presented as percentage of a full lysis control (A-431 treated with Triton X-100). The dotted line represents damage from uninfected A-431 cells. Albumin was always used at a 5 mg/mL concentration. One-way ANOVA was used to calculate statistically significant differences of c_WT, *hap5*Δ and c_ *hap5*Δ *vs*. WT. Statistical significance is indicated as: ****, $p \leq 0.0001$; ns, not significant.

ferrous sulfate (Fig 6d). The increased iron availability in the form of ferrous sulfate, similar to albumin, enabled *C. glabrata* to cause host cell damage. More precisely, supplementation of 2.3 mg/L of ferrous sulfate resulted in a similar damage level as the one observed in RPMI with albumin and no additional iron added (Fig 6d).

Moreover, during infection in the presence of albumin, *C. glabrata* drastically changed the expression of genes involved in iron uptake (*FTR1* and *SIT1*), intracellular iron distribution and consumption (*FTH1*, *HMX1*) and virulence (*EPA1*, *YPS2*, *AUS1*) (Fig 6e). The *hap5*Δ mutant showed significantly lower expression levels of these genes as compared to the WT at 24 hpi (Fig 6e).

## *C. glabrata* infection induces glucose starvation and ER-stress in vaginal epithelial cells in the presence of albumin

To shed light on the mechanisms by which *C. glabrata* induces epithelial damage, we analyzed the transcriptional responses of the epithelial cells during infection in the presence and absence of albumin (a PCA plot of all analyzed samples is shown in S2C Fig).

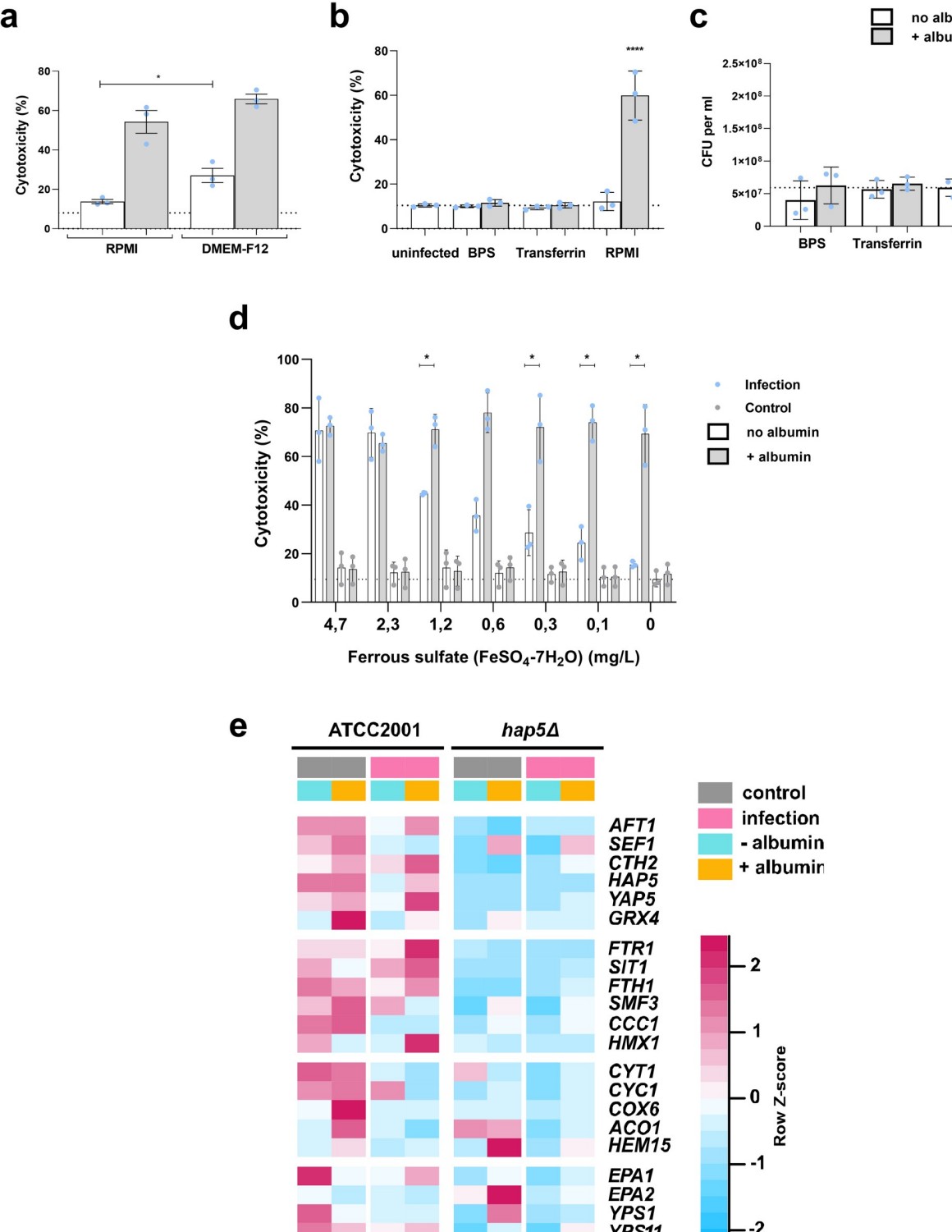

**Fig 6. Role of iron in *C. glabrata* infection. (a)** Damage of A-431 cells infected with *C. glabrata* in RPMI or DMEM-F12 medium, with or without albumin. **(b)** Damage of A-431 cells infected with *C. glabrata* with or without albumin with or without 2.5 mg/mL transferrin or 50 μM Bathophenanthrolinedisulfonic acid disodium salt (BPS). **(c)** *C. glabrata* growth 24 h post infection of A-431 cells with or without albumin with or without 2.5 mg/mL transferrin or 50 μM BPS. **(d)** Damage of A-431 cells infected with *C. glabrata* with or without albumin in RPMI supplemented with ferrous sulfate (concentration range 0–4.7 mg/L), **(e)** Comparison of the expression of iron-related genes between *C. glabrata* wild type (WT)

and *hap5Δ* during infection or in medium only in the presence or absence of albumin at 24 hpi. Shown are qRT-PCR data as fold change of values obtained after incubation for 0.5 h in RPMI medium without albumin and host cells. Z-score was used to scale the values within one row (one gene). All values are presented as mean ± SD. Damage was recorded by measuring the lactate dehydrogenase activity in the supernatant and presented as percentage of a full lysis control (A-431 treated with Triton X-100). The dotted line represents damage from uninfected A-431 cells. Albumin was always used at a 5 mg/mL concentration. One-way ANOVA was used to calculate statistically significant differences. Statistical significance is indicated as: *, $p \leq 0.05$; **, $p \leq 0.01$; ***, $p \leq 0.001$; ns–not significant.

Transcriptional analysis indicated infection-specific up-regulation of genes associated with glucose starvation, such as *SESN2* that encodes for stress-inducible metabolic regulator Sestrin2 (S2D Fig). Therefore, glucose consumption may contribute to epithelial cell death during epithelial infection in the presence of albumin.

Furthermore, GO-term enrichment analysis revealed that the epithelial cells experience ER stress and mount an unfolded protein response (UPR) (Fig 7a). We next analyzed the expression pattern of selected genes potentially relevant for our observations and hypothesis (Fig 7b). An infection-specific upregulation of several genes related to ER-stress was observed. These genes encode sensors and transcription factors that mediate induction of proinflammatory responses and apoptosis during viral and bacterial infections [34] or regulate plasma membrane cholesterol and intracellular cholesterol homeostasis, such as StAR-related lipid transfer protein 5 (StARD5) [35].

In line with the observed and known importance of iron during infection, host cells upregulated the gene *LTF* encoding lactoferrin, a peptide which has both antimicrobial activity and iron binding capacities and can thereby deprive pathogens from iron [36]. Expression of gene *LL37*, encoding for an antimicrobial peptide, was up-regulated by albumin both during infection and in uninfected conditions. Finally, the upregulation of the gene coding for IL-1α is in accordance with the observed level of released lactate dehydrogenase (LDH) at 24 hpi, as IL-1α is a major alarmin that induces proinflammatory responses upon epithelial cell damage [37–40].

To investigate the consequence of the observed ER-stress response on infection, we preinduced ER-stress in A-431 cells, by pre-incubating them with tunicamycin, before *C. glabrata* infection in the presence or absence of albumin. Interestingly, ER stress induction prior to infection conferred resistance to *C. glabrata*-induced damage in the presence of albumin (Fig 7c). This suggests that the ER stress pathway and UPR could be protective for the epithelial cells, as previously described for viral infections [34].

## Albumin-augmented damage potential under physiologically relevant conditions

To increase the physiological relevance of our findings, we considered aspects that are characteristic for the vaginal mucosa and VVC, such as lactobacilli, pH, and glucose levels. *Lactobacillus* spp. represent the dominant bacterial species of vaginal microbiota and a variety of studies highlight their protective role in protecting against infections by *Candida* species [41–43]. Compared to the control infection conditions (S3A Fig), colonization with *L. rhamnosus*, prevented the increased epithelial damage in the presence of albumin (S3B Fig), similarly to how *L. rhamnosus* prevents damage induced by the highly pathogenic species *C. albicans* [44].

The normal pH range in the vagina is between 3.8 and 4.5, and since a low pH drastically influences biochemical reactions, we investigated whether albumin can still enhance *C. glabrata* pathogenic potential at a pH of 4. At this low pH we observed a similar increase of damage potential compared to the pH value of 7–7.4 (S3C Fig), indicating that the phenomenon of increased epithelial damage in the presence of albumin could potentially occur in the acidic environment of the vaginal mucosa.

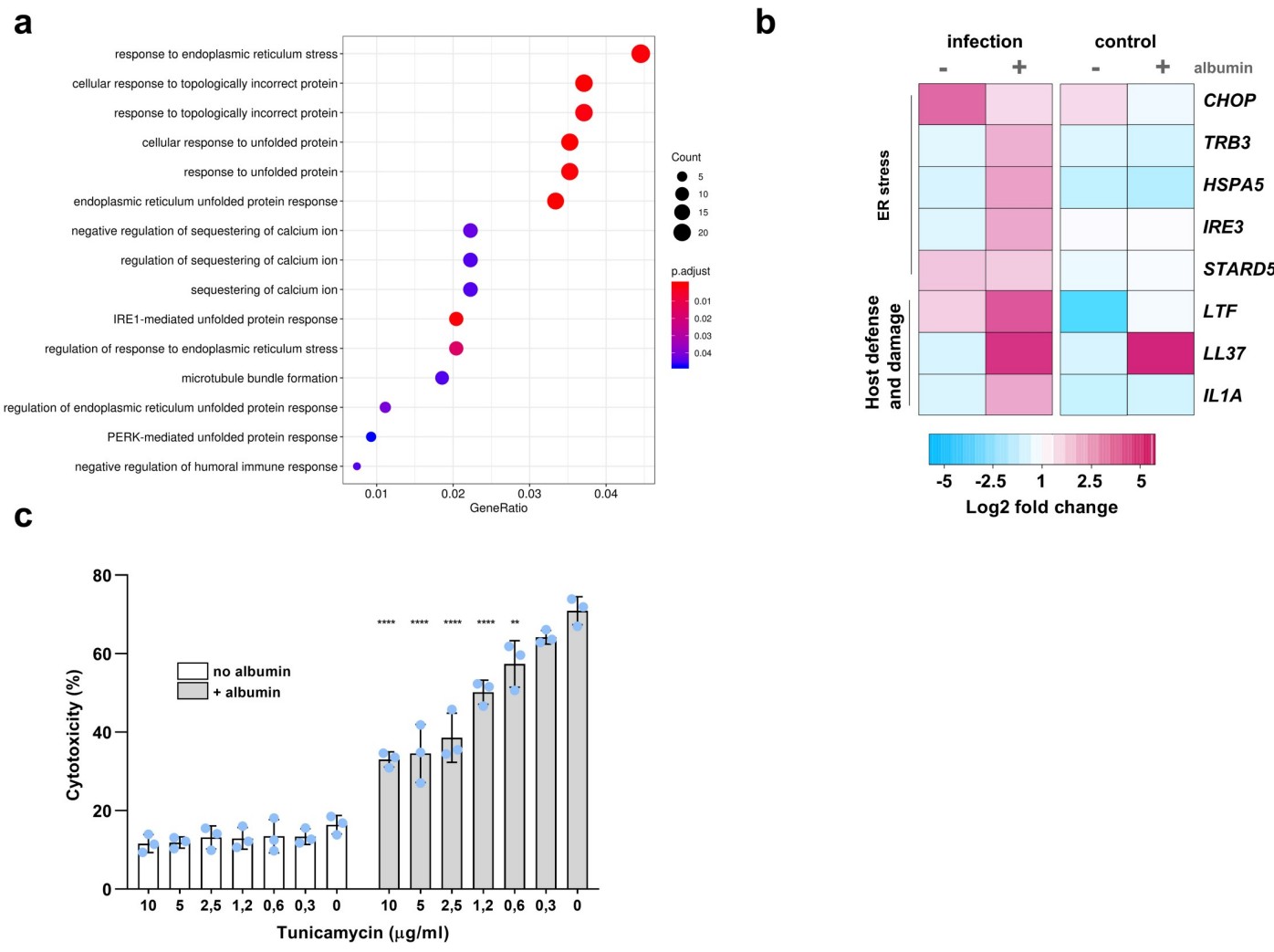

**Fig 7. Host transcriptional response to *C. glabrata* infection in the presence of albumin. (a)** Gene-Ontology (GO) term enrichment analysis for up-regulated genes (category "Biological Processes") at 24 hours post-infection (hpi). Only significant ($p < 0.05$) GO enrichments are shown (none was found to be significant at 3 hpi). Counts correspond to the number of genes assigned to each GO category. GeneRatio corresponds to the ratio between the number of genes enriched in a given category and "counts". Adjustment of *p*-values was performed by the Benjamini-Hochberg procedure. **(b)** Expression of selected relevant host genes at 3 and 24 hpi and 24 h control (24 c) with or without albumin, derived from RNA-Seq data (see Material and Methods). Presented values are expressed as $\log_2$ fold changes of expression compared to host cells in medium only at the time point 0. **(c)** Damage of tunicamycin (TM)-pretreated A-431 cells infected with *C. glabrata* with or without albumin. All values are presented as mean ± SD. Damage was recorded by measuring the lactate dehydrogenase activity in the supernatant and presented as percentage of a full lysis control (A-431 treated with Triton X-100). The dotted line represents damage from uninfected A-431. Albumin was always used at a 5 mg/mL concentration. One-way ANOVA was used to compare the extents of damage after TM pre-treatment (0.3–10 µg/mL) to the damage level without TM pre-treatment (0). Statistical significance is indicated as: **, $p \leq 0.01$; ****, $p \leq 0.0001$.

Finally, glucose is an important factor during infection as: (i) women with diabetes tend to have higher prevalence of *C. glabrata* vaginal infection [45–47] and (ii) our data indicated that epithelial cells induce glucose starvation response upon the infection. Therefore, we investigated the role of albumin in the absence of glucose, and observed no induction of damage in the presence of albumin (S3D Fig), which was associated with significantly reduced *C. glabrata* growth (S3E Fig). Similarly, an increased glucose availability (20 mM) also reduced the epithelial damage, probably by preventing host cell glucose starvation thereby improving the epithelial cell survival (S3D Fig).

In summary, we demonstrated that albumin enhances *C. glabrata* pathogenic potential during interaction with vaginal epithelial cells. This suggests that albumin is as a key player in the pathogenesis of VVC (Fig 8).

## Discussion

Although *C. glabrata* frequently causes vulvovaginal candidiasis (VVC), this fungus is almost avirulent in mice and does not cause damage to human epithelial cells *in vitro* under typical infection conditions [9]. In this study, we found that human and bovine, but not mouse albumin activates the pathogenic potential of *C. glabrata* during interaction with human vaginal epithelial cells.

Epithelial damage caused by *C. glabrata* in the presence of albumin reached comparable levels of damage caused by the highly damaging species *C. albicans* [9,48–50]. Importantly, these effects relied on albumin uptake and proteolytical processing by epithelial cells.

Vaginal epithelial cells up-regulated mRNA expression of albumin receptor genes and genes encoding proteins involved in albumin binding and uptake, in the presence of albumin. We demonstrated albumin uptake by vaginal epithelial cells. This uptake is possibly mediated *via* macropinocytosis, as inhibition of this type of endocytosis prevented the damage by *C. glabrata* in the presence of albumin. This is in agreement with a previous study showing that albumin uptake by alveolar epithelial cells (A549) was mediated by macropinocytosis [51].

Several studies reported that albumin acts as a "mother protein" that generates different peptides and biomarkers, depending on the examined body fluid [52–55]. These bioactive peptides can be released into the extracellular space upon host cell death [56]. *C. glabrata*, may

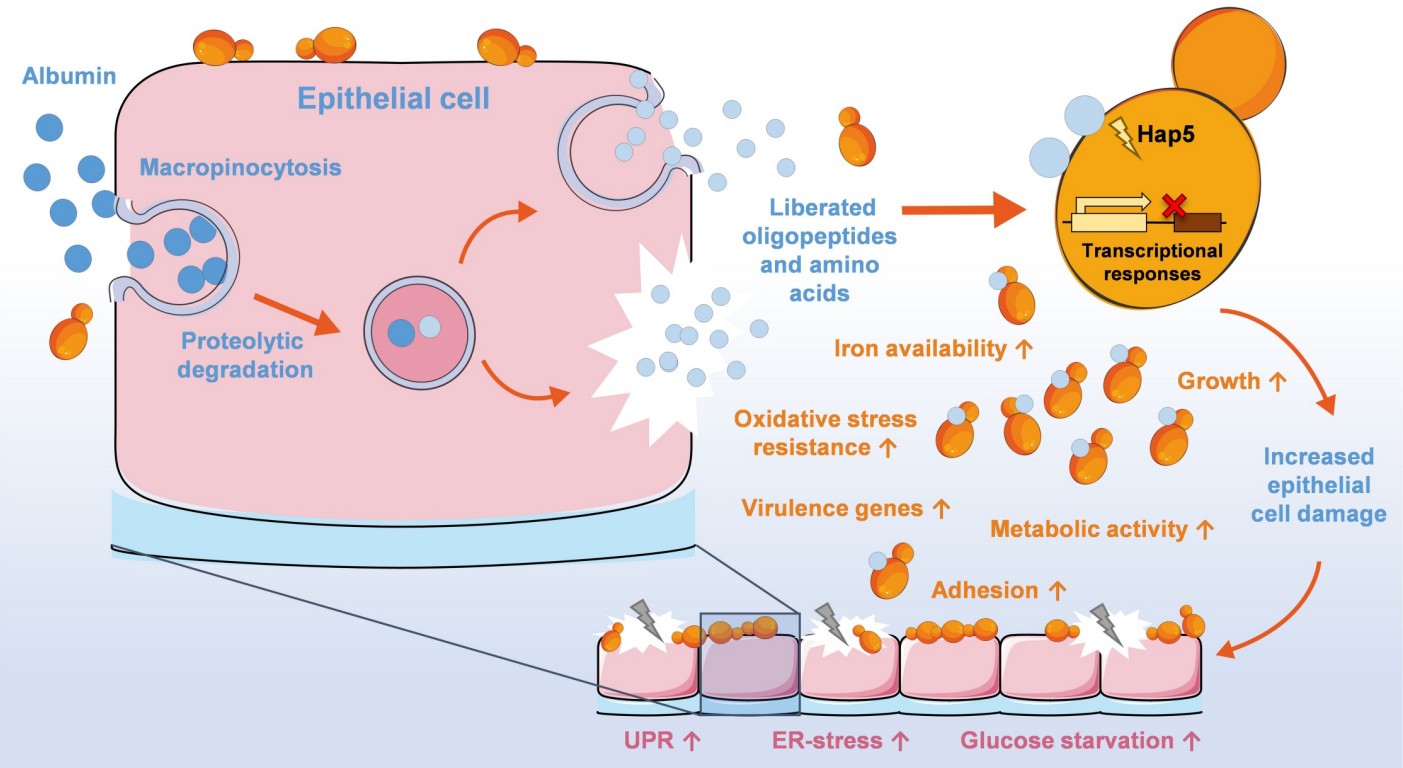

**Fig 8. Schematic model of the main findings of current study.** Schematic model was created with images adapted from Servier Medical Art by Servier. Original images are licensed under a Creative Commons Attribution 3.0 Unported License.

encounter these peptides, which can modulate its growth and pathogenic potential. The importance of proteolytic albumin processing into bioactive peptides was further suggested as protease inhibitors prevented *C. glabrata*-induced damage in the presence of albumin.

We propose that these processes enhance *C. glabrata* pathogenicity by a combination of beneficial effects for the fungus: increased access to iron, accelerated growth, increased adhesion, and increased fitness, potentially associated with the expression of distinct virulence associated genes.

Albumin is the most abundant host heme-scavenging protein, thus limiting the major source of iron for microbial pathogens [21,57,58]. Surprisingly, in the case of *C. albicans*, albumin promoted, rather than prevented, heme utilization [21]. Our findings also support that albumin, or most likely its degradation products, support *C. glabrata* to overcome iron restriction in a similar manner. Alternatively albumin may provide another unknown limiting factor crucial for increased virulence potential. Nevertheless, the addition of iron-chelating factors strongly inhibited—whereas iron supplementation increased—*C. glabrata* mediated damage in the presence of albumin. This strongly suggests that iron access and homeostasis is central to the enhanced damage capacity in the presence of albumin. Iron metabolism and uptake were also shown to be crucial for most human fungal pathogens, including *C. albicans* [59], *C. parapsilosis* [60], *Aspergillus fumigatus* [61,62], and Mucorales species [63], but the role of albumin in this context was not yet investigated.

By screening a *C. glabrata* gene deletion mutant library of over 600 strains [29], we identified Hap5, an essential transcriptional activator for iron homeostasis [30–32], as a key player in the albumin induced pathogenic potential of *C. glabrata*. Furthermore, the comparison of WT and *hap5Δ* transcriptional responses to infection in the presence of albumin demonstrates a failure to induce genes required for iron uptake, intracellular iron distribution and consumption in *hap5Δ*. While these genes have been associated with iron starvation [32], they are also induced during iron excess [64] and may also be linked to further cellular processes. However, the differential expression of these genes further underscores that iron homeostasis is crucial for albumin-augmented damage in our *in vitro* infection model.

Increased access to iron and Hap5-mediated iron homeostasis in the presence of albumin, or the proteolytic products of it, leads to accelerated proliferation of *C. glabrata*. This at least partially explains the increased damage. However, our data show that further fungal attributes must contribute to the albumin-mediated damage. First, we monitored increased adhesion, potentially mediated by the increased expression of adhesins (e.g. genes coding Awp proteins), which allow close contact of *C. glabrata* cells to epithelial cells. Second, due to the increased access to iron the fungal cells are likely more physiologically active and show increased metabolic and mitochondrial activities associated with the production of ROS. *C. glabrata* transcriptional response revealed up-regulation of oxidative stress responses. At first glance, this seems to be a contradiction, because albumin exhibits antioxidant properties by scavenging free radicals [65,66] and, therefore, its presence could be beneficial for the fungus to cope with host defense mechanisms causing oxidative stress. However, due to the proteolytic digest and uptake of albumin, this potential scavenging effect of albumin may get lost. Our data also show that the presence of albumin does not induce ROS production by epithelial cells. Therefore, we can exclude that the oxidative stress response of *C. glabrata* cells is triggered by the epithelial cells. Instead, we propose that the up-regulation of oxidative stress resistance genes in *C. glabrata* may rather be an intrinsic mechanism to compensate for ROS production associated with the increased metabolic rates and mitochondrial activity, facilitated by increased iron availability. Similarly, increased oxidative stress resistance as a consequence of increased iron availability has also shown been shown in bacterial pathogens such as *Clostridioides difficile* [67]. *In vivo*, such

an increased oxidative stress resistance could be beneficial for fungal cells challenged by phagocytic innate immune cells attracted by epithelial damage.

Higher fungal metabolic activity is associated with increased glucose consumption which causes glucose starvation of epithelial cells. The infection-specific up-regulation of epithelial genes associated with glucose starvation confirmed this. Increased glucose availability during infection rescued epithelial cell survival in the presence of albumin. The importance of glucose for *C. glabrata* pathogenicity was underscored by infections in the absence of glucose, where *C. glabrata* proliferation rates were significantly lower and no tissue damage in the presence of albumin was induced. Therefore, glucose availability seems to plays an important role in albumin-enhanced *C. glabrata* pathogenicity, which is especially relevant when taking into account the high incidence of VVC caused by *C. glabrata* in diabetes mellitus patients [45–47].

Increased pathogenic potential may also be associated with the expression of virulence associated genes such as *YPS1* and *AUS1*. The *YPS* gene family encodes glycosylphosphatidylinositol-linked aspartyl proteases and is required for *C. glabrata* virulence [25]. *AUS1* encodes a sterol importer. Albumin mediates cholesterol efflux from cultured endothelial cells [68]. The fact that *C. glabrata* highly up-regulated the gene encoding for Aus1 during infection once albumin is present may indicate that cholesterol efflux also may occur in our model. Of note, infection with the *C. glabrata hap5Δ* mutant caused lower levels of *AUS1* gene expression as compared to infections with the WT in the presence of albumin, supporting that iron is required for this process [69]. *C. glabrata* may use host cholesterol to sustain its growing population with sterols for the synthesis of new membranes [70,71]. The augmented growth was contact-independent, although slightly lower replication rates were observed in transwells. In contrast, the augmented damage potential was only observed when *C. glabrata* was in direct contact with the host cells. Therefore, we could speculate that cholesterol uptake mechanisms may scavenge sterols from the host membrane, as shown previously for *Pneumocystis carinii* [72,73]. The host *STARD5* gene encoding an ER stress-responsive protein that regulates plasma membrane cholesterol and intracellular cholesterol homeostasis [35], was up-regulated upon *C. glabrata* infection in the presence of albumin. The scavenging of host membrane sterols may compromise membrane integrity and could be a mechanism by which *C. glabrata* damages host cells. Host sterol scavenging could also be a strategy of immune evasion. While fungal ergosterol can stimulate macrophages [74] and trigger pyroptosis [75], the integration of cholesterol in the membrane would display as a "self" molecule, therefore remaining undetectable by the host immune system. This is in line with the general infection strategy of *C. glabrata* to evade immune recognition and secure persistence [76]. Further studies are needed to confirm this hypothesis.

These observations can explain the fungal fitness and properties leading to epithelial damage. We further propose a combination of at least two mechanisms by which *C. glabrata* causes damage.

First, transcriptional responses of the epithelial cells suggested a dysfunctional secretory pathway. Only during infection in the presence of albumin, epithelial cells induced genes associated with ER-stress and UPR. Protein-folding within the ER is influenced by different stressors, resulting in the accumulation of unfolded or misfolded proteins, which can lead to cytotoxicity [34,77]. Once the UPR is activated, this signaling involves genes coding for inositol-requiring enzyme 1 (*IRE1*), protein kinase R (PKR)-like ER kinase (*PERK*), and immunoglobulin heavy chain-binding protein (*BiP/GRP78*) [78,79]. The corresponding genes of these proteins were all significantly up-regulated upon *C. glabrata* infection in the presence of albumin. The induction of ER-stress is induced in many bacterial and viral infections [34], but little is known about the role of ER-stress in pathogenesis of *Candida* infections. For example, ER stress can activate proinflammatory signals and regulate immune responses (reviewed in [34]).

Surprisingly, we observed that the induction of ER-stress responses conferred resistance to albumin-induced *C. glabrata* pathogenicity. A possible explanation may be that the induced UPR response affects albumin uptake and processing, which is crucial for *C. glabrata* mediated damage in this scenario. For example, *CHOP* was up-regulated in our dataset, a gene that is coding for a protein that mediates ER-stress-induced apoptosis and proinflammatory cytokine production [80].

Second, as discussed above, fast growing *C. glabrata* cells attached to epithelial cells likely caused glucose starvation in epithelial cells. Excessive consumption has also been claimed as a mechanism causing host cell death during interaction of *C. albicans* and macrophages [81]. Therefore, glucose consumption may similarly contribute to vaginal epithelial cells death during epithelial infection in the presence of albumin.

Interestingly, a study from Ding *et al*. (2016) showed that glucose starvation and UPR can be connected [82]. In this study it was reported that UPR induces the above-mentioned Sestrin2 that protects cells from glucose starvation-induced cell death. This further points out that metabolic stress might be at the core of the observed host cytotoxicity.

Finally, the observation that human and bovine, but not mouse albumin augmented *C. glabrata* pathogenicity is intriguing and could potentially help to explain why *C. glabrata* is avirulent in most mouse models [8], unless hypervirulence is induced by inactivation of virulence-moderating gene [83]. The underlying mechanism why murine albumin was unable to enhance *C. glabrata* pathogenicity is unclear and it can only be speculated that this may be due to differences in protein sequence, structural differences that impact receptor binding [84] and function, or uptake and processing by human cells, compared to human and bovine albumin. However, our data clearly show that the addition of a single host protein can dramatically alter the damage potential of *C. glabrata*.

In summary, we suggest that albumin could play a key role in the pathogenesis of VVC, by boosting *C. glabrata* virulence, leading to epithelial cell ER-stress and damage. Albumin concentrations are highly variable in different physiological and pathological conditions. For example, serum albumin levels are drastically decreased during inflammation and malnutrition, which was associated with the poor outcome in critically ill patients [85–88]. Albumin levels in saliva were shown to be significantly increased during HIV infection and in elderly people [12], potentially explaining the increased oropharyngeal colonization with *C. glabrata* in older patients [89]. Similarly, albumin concentrations were found to be increased during infection in early pregnancy [90]. Monitoring albumin levels in VVC patients could be of importance to predict the infection outcome, but no study so far measured this parameter.

## Material and methods

### Strains and culture conditions

The strains used in this study were *Candida glabrata* ATCC 2001 (obtained from American Type Culture Collection; ATCC), *C. albicans* SC5314 [91], *C. glabrata* deletion-mutant strains described in [29] and the corresponding isogenic wild type (c_WT; triple auxotrophic *his3Δ leu2Δ trp1Δ* ATCC 2001), and *C. glabrata hap5Δ* strain (*hap5Δ*) described in [32]. *C. glabrata* clinical isolates were kindly provided by Dr. Oliver Bader (Institute for Medical Microbiology, University Medical Center Göttingen, Göttingen, Germany). *Lactobacillus rhamnosus* ATCC 7469 was used for colonization experiment.

For overnight cultures, single colonies were picked from Yeast Peptone Dextrose (YPD) agar plates and grown for 16 h in liquid YPD medium in an orbital shaker at 180 rpm at 30 ˚C (*C. albicans*) or 37 ˚C (*C. glabrata*). Yeast cells were then harvested by centrifugation (20 000 g, 1 min), washed twice with phosphate-buffered saline (PBS), and adjusted to $2\times10^7$ yeast cells

per milliliter (yeast/mL) prior to final dilution in the respective assay. *L. rhamnosus* was grown in MRS (deMan, Rogosa and Sharpe) medium for 48h, at 37˚C, 5% $CO_2$, and 1% $O_2$ under static conditions statically.

## Chemicals

Human albumin was obtained from Sanquin Plasma Products B.V. and tested in the concentration of 5 mg/mL. For S1 Fig, human albumin obtained from Bio & Sell catalog number HSA.FV.0025 and Sigma, catalog number A6909 were used. Bovine serum albumin (BSA) was obtained from Serva and used for mutant library screening in the same concentration. Murine serum albumin was obtained from Sigma (catalog number A3139), as well as from Innovative Research (catalog number IMSALB100MG) and Abcam (catalog number ab183228). Apo-transferrin (Sigma-Aldrich) and Bathophenanthrolinedisulfonic acid disodium salt (BPS; Alfa Aesar) were tested as iron-chelators. Human serum albumin conjugated with Alexa Fluor 647 (Jackson ImmunoResearch) was used for visualization of albumin uptake (1 mg/mL). Protease inhibitor cocktail was obtained from Roche (Roche cOmplete Mini tablets). As endocytosis inhibitors, the following compounds were used at the indicated concentrations (all obtained from Sigma-Aldrich): amiloride 0.5 mM, genistein 125 μM, simvastatin 10 μM, colchicine 50 μM, methy-β-cyclodextrin 3 mM, nocodazole 66 μM, and cytochalasin D 0.5 and 10 μM. Oxidative stress was induced by hydrogen peroxide ($H_2O_2$; Roth). ER-stress was induced by tunicamycin (Sigma).

## Fungal growth analysis

*Candida* cells were adjusted to $1 \times 10^5$ cells/mL either in YPD or in RPMI 1640 (RPMI) medium with or without albumin. Growth was monitored in 96-well-plates in a volume of 200 μL by measuring the absorbance at 600 nm every 30 min for 40–50 hours at 37˚C in a microplate reader (Tecan). Prior to each measurement, plates were orbitally shaken for ten seconds followed by ten seconds waiting time.

## *In vitro* vaginal epithelial infection model

A-431 epithelial cells (ECs, Deutsche Sammlung von Mikroorganismen und Zellkulturen DSMZ no. ACC 91) were used to mimic vaginal mucosal infection [92,93]. A-431 ECs were cultured in RPMI medium supplemented with 10% heat-inactivated fetal bovine serum (FBS) at 37˚C and 5% $CO_2$. For the infection, 200μL or 1mL of $1 \times 10^5$ ECs per mL were seeded in 96-well-plates or 24-well-plates respectively and cultured at 37˚C and 5% $CO_2$ for 2 days. Spent medium was replaced with fresh standard RPMI (Gibco, catalog number 21875034), RPMI without glucose (Gibco, catalog number 11879020), RPMI supplemented with 20mM glucose, or DMEM-F12 without FBS and with or without 5 mg/mL albumin. ECs were infected with the desired *Candida* strain at a multiplicity of infection (MOI) of 1 ($1 \times 10^5$ yeast/ mL), unless stated otherwise. Infected ECs were incubated at 37˚C and 5% $CO_2$ for 24 h for cytotoxicity assays or 1 h, for adhesion assay. For transwell assays, ECs were grown for 2 days in 24-well plates. For infection, used medium was aspirated and replaced with 750 μL fresh RPMI medium or RPMI medium containing albumin. Transwell inserts (polycarbonate membrane inserts with 0.4 μm pore size; Corning), loaded with 250 μL *C. glabrata* suspension ($1 \times 10^5$ yeast cells in total for an MOI of 1), were placed in the wells. For endocytosis inhibition assays, ECs were incubated with the inhibitors in RPMI without FBS for 1 hour prior to infection. For investigating the effect of increased iron availability, infection experiments were performed using RPMI supplemented with Iron (II) sulfate heptahydrate ($FeSO_4 \cdot 7H_2O$, Sigma) in the concentration range 0–4.7 mg/L. For investigating the impact of colonization by

lactobacilli, ECs were colonized with 100 µL of *L. rhamnosus* (OD 0.1 in RPMI) for 18 h prior to infection. Subsequently the infection was performed as described above.

### Adhesion assay

For adhesion assays, ECs were seeded on sterile glass cover slips (Ø 12 mm; Carolina), cultured and infected as described above. At 1 hpi, infected ECs were washed with PBS to remove non-adherent *C. glabrata*. ECs and adherent fungal cells were fixed in 4% Histofix (Carl Roth), stained with Alexa Fluor 647 conjugate of succinylated concanavalin A (ConA; Invitrogen), and visualized using a fluorescent microscope (Leica DM5500B, 40× magnification). At least 100 adherent yeast cells were counted for each sample, and adhesion was calculated based on the average of yeast cells counted in each microscopic frame of a defined area. Adhesion values are expressed as a percentage of adherent yeasts, compared to the initial inoculum [49].

### Epithelial damage assay

Epithelial damage was determined at 24 hpi by measuring the activity of the cytoplasmic enzyme lactate dehydrogenase (LDH) in the supernatant, which is released as a consequence of loss of membrane integrity upon necrotic cell damage [94], using a Cytotoxicity Detection Kit (Roche) according to the manufacturer's protocol. All measurements were performed in triplicates, and values were plotted as % of full lysis (induced by the addition of 0.25% Triton X-100 to uninfected ECs). For mutant screening, LDH release was determined after 24 hpi, in both BSA and non-BSA infection conditions and expressed as relative to wild type-induced damage (%). Obtained data was filtered by excluding strains that showed less than 80% of the damage capacity of the *C. glabrata* collection wild type (c_WT; ATCC 2001 triple auxotroph *his3Δ leu2Δ trp1Δ*) in albumin-free medium and more than 30% of c_WT damage capacity in albumin-containing medium at 24 hpi. The resulting strains were analyzed again to confirm the damage deficiency in the presence of human albumin and BSA, as well as to compare the damage between c_WT and WT used in other experiments (ATCC 2001); and *hap5Δ* mutants generated in c_WT and WT background.

### Colony-forming unit (CFU) counting

Well contents were plated on YPD agar plates and the number of single colonies was counted after 24 h incubation at 37˚C. For each sample, well content was transferred to the new tube and the wells were additionally washed three times with PBS. All washes were collected in the same tube to ensure the transfer of all yeast cells. For transwell experiments, transwell inserts with fungal growth were placed in 50 mL centrifuge tubes containing 10 mL PBS and vortexed to detach all yeast cells. In order to achieve countable numbers of single colonies, all samples were diluted $1:10^2$–$1:10^4$ prior to plating. Values are expressed as CFU/mL of the original well content, after multiplying the number of counted colonies by the corresponding dilution factor.

### Fluorescence microscopy

ECs were seeded in 96-well plates ($2×10^4$ cells/well) and infected with *Candida* cells ($2×10^4$ cells/well) in the presence of 1 mg/mL human albumin-Alexa Fluor 647. At 6 hpi, ECs were washed, fixed with 4% Histofix and visualized using a Zeiss Cell Discoverer microscope (Carl Zeiss) with fluorescence settings at 650/665 nm.

## ROS production by ECs

ECs cells were stained with 2',7'–dichlorofluorescin diacetate (DCFDA; fluorogenic dye that measures ROS activity within the cell) using DCFDA / H2DCFDA—Cellular ROS Assay Kit (Abcam), according to the manufacturer's instructions. Stained ECs were infected with *C. glabrata* in RPMI with or without 5 mg/mL albumin and the fluorescence intensity was measured over 24 hours (485/535 nm). Uninfected cells in RPMI with or without 5 mg/mL albumin were included as control.

## ER-stress induction

Confluent A-431 cells were pre-treated with different concentrations of tunicamycin (0.3–10 μg/mL) for 4 h to induce ER-stress. Subsequently, the medium containing tunicamycin was removed, cells were washed and incubated for another 24 h in RPMI with FBS. On the following day, A-431 cells were infected with *C. glabrata* in RPMI with or without 5 mg/mL albumin and the LDH release was measured at 24 hpi.

## RNA isolation

On the day of infection, the media of A-431 cells was exchanged with RPMI without FBS and incubated for 30 min in order to allow cells to adjust to the change of medium (this was considered as time point 0 in bioinformatical analyses). ECs were then infected with *Candida* cells ($1 \times 10^6$ yeast/mL in RPMI without FBS) and incubated at 37˚C, 5% $CO_2$. Samples for RNA isolation were collected at 3- and 24 hours post-infection (hpi) by removing the well content and adding RNeasy Lysis (RLT) buffer (Qiagen), containing 1% β-mercaptoethanol (Roth). Cells were detached using a cell scraper, frozen in liquid nitrogen, and stored at -80˚C. As controls, *C. glabrata* WT or *hap5Δ* cells [32] alone and ECs alone were incubated for 30 min (0 h control) and 24 hours (24 h control) and samples for RNA isolation were collected as described above. Fungal and human RNA were isolated as described previously [9,49]. Briefly, collected samples were defrosted on ice and centrifuged for 10 min (20 000 g, 4˚C). The supernatant was used to isolate human RNA (RNeasy Mini Kit, Qiagen), while fungal RNA was isolated from the pellet, using a freezing-thawing method [49]. RNA was quantified using a Nanodrop ND-1000 Spectrophotometer (VWR International) and its quality checked by running an RNA Nano Chip (Agilent Technologies) on a Bioanalyzer (Agilent 2100 Bioanalyzer) using the Agilent RNA 6000 Nano kit. For RNA-Seq analysis, we used a previously validated approach of pooling corresponding fungal and human RNA samples in a 2:3 quantitative ratio by weight for further library preparation and sequencing [9,95].

## RT-qPCR

For RT-qPCR, 500 ng RNA was treated with DNase (Fermentas) for 30 min at 37˚C and reverse transcribed into cDNA using 0.5 μg Oligo(dT)12-18 Primer, 200 U Superscript III Reverse Transcriptase and 40 U RNaseOUT Recombinant RNase Inhibitor (Thermo Fischer Scientific). cDNA (100 ng) was used for RT-qPCR assay, with GoTaq qPCR mastermix (Promega) in a C1000 thermocycler (BioRad, CFX96 Realtime system). The expression levels were normalized to the housekeeping gene *ACT1* and expressed as relative to the expression of the target gene at 0 time point ($\log_2$ fold change). Gene expression analysis was carried out in triplicates for each sample and primer pair. All the primers used can be found in Table 1.

**Table 1. List of primers used in this study.**

| Gene | SEQUENCE 5'-3' | |
|------|-----|-----|
| | FW | RV |
| AFT1 | CAAGACATGACGTTGGAGATGA | CCCGCGTTTACAGGATAGTCA |
| SEF1 | TCTGCCCGAGATACCGAAGA | AGACGACTGTTGGCTGATGG |
| CHT2 | TTGGTTACTGCCCATACGGT | CCATCAGGGTTTGCAGTATAGGT |
| HAP5 | GAAGGCGTGCGAAGTGTTTA | CTCATCTGCAGGGCCTCAG |
| YAP5 | CGGGCCTATCAACAGAATCCA | AGCATTTTGTACACTTGGATGGA |
| GRX4 | CTGGCCTACGTTCCCTCAAT | ACTCGTATGGTGTGTTGGAAGA |
| FTR1 | CGAAGAAGAACACGACCAGGA | GTCGGAACTGGTAGTGGTAAC |
| SIT1 | CCTGGTCTTCTGTGCACCTC | TGGCACCCTTGAACCATTCG |
| FTH1 | GGTTACATTCCTGGGTTGCC | GCTATCCTTTGAAACACGCTGT |
| SMF3 | TGATGTTGACGAGAGCACAC | CCAAACAAAGACAGCTGCTCC |
| CCC1 | GGTGCCCTTGGTGCCATATT | TAGCCGAACCAGAACAACGT |
| HMX1 | TCGGCCAGCTGAACAAAGAG | GGCTCCTGTCCTTGTTGTAGA |
| ACO1 | TGCTCCAGGTAAGAACGTCAC | AGCAGAACCGTACTTGAACCA |
| CYC1 | TACACCGACGCCAACATCAA | AGACCACCGAAAGCCATCTT |
| CYT1 | GCCAGAACACGACGAGAGAA | TCTTGGCTTTGGTGGGTTGA |
| COX6 | GCTGTCATCGAGAAGGCTTTG | AGCCTTGTATTGGTCCTCATTT |
| HEM15 | GCTTTCACATCTGACCACATCG | GGACTTCCGTTCAATGATTCACA |
| EPA1 | ACAGCGAGGAACACAATAGCA | AGCAAAAGTTGAGTGTATCCCA |
| EPA2 | GGCAACAACGGCAATGGTAA | GCAGCCCTAAATCCTTCACCT |
| YPS1 | CGTTCTCGTACCTCTGCTACA | ACCAAAAGACCAGCGACGAA |
| YPS11 | AGTTGTCGTCAGCACCATCC | AGTTTGTGCTGCCGTTTGAG |
| AUS1 | GTGGGGATTTCGACGAGTAT | CGAAGGAACTACCGTGAATTT |
| ACT1 | CTGTCTGGATCGGTGGTTCT | GATGGACCACTTTCGTCGTA |

## Oxidative stress resistance

*C. glabrata* cells were used to infect ECs with or without albumin as described above. After 3 h, fungal cells were collected, washed and resuspended in RPMI with or without 10 mM $H_2O_2$ at 37°C for an hour. The samples were diluted $1:10^4$ and plated on YPD agar plates to determine the number of CFUs.

## RNA-Seq library preparation and sequencing

Library preparation for RNA-Seq was performed at the Genomics Unit of the Centre for Genomic Regulation (Barcelona, Spain) with the TruSeq Stranded mRNA Sample Prep Kit v2 from Illumina according to the manufacturer's instructions. One µg of total RNA was used for poly(A)-mRNA selection using poly-T oligo attached magnetic beads. Samples were then fragmented to ~300bp and subsequently, cDNA was synthesized using reverse transcriptase (SuperScript II, Invitrogen) and random primers. The second strand of the cDNA incorporated dUTP in place of dTTP. Double-stranded DNA was further used for library preparation. It was subjected to A-tailing and ligation of the barcoded Truseq adapters. All purification steps were done using AMPure XP beads (Agencourt). Library amplification was performed by PCR on the size selected fragments using the primer cocktail supplied in the kit. Final libraries were analyzed using Agilent DNA 1000 chip (Agilent) to estimate the quantity and check fragment size distribution and were then quantified by qPCR using the KAPA Library Quantification Kit (Kapa Biosystems) before amplification with Illumina's cBot. All libraries were sequenced in two batches with 2x50 read length

using Illumina HiSeq2500 at the Genomics Unit of the Centre for Genomic Regulation (Barcelona, Spain).

## RNA-Seq data analysis

FastQC v. 0.11.6 (https://www.bioinformatics.babraham.ac.uk/projects/fastqc/) and Multiqc v. 1.0 [96] were used to perform quality control of raw sequencing data. For read mapping and quantification, we used splice-junction sensitive read mapper STAR v. 2.5.2b [97] using basic two-pass mode and default parameters. Considering that samples contained RNA from both the host and the pathogen, the data were mapped to concatenated human and *C. glabrata* reference genomes. For human data, we used the primary genome assembly GRCh38 and genome annotations from Ensembl database release 89 (last accessed on 8 of August 2017) [98]. Reference genome and genome annotation of *C. glabrata* CBS138 was obtained from Candida Genome Database (CGD, last accessed on 17 of August 2017) [99]. GFF genome annotation file of *C. glabrata* was converted to GTF format using gffread utility v. 0.9.8 [100]. Differential gene expression analysis was performed using the Bioconductor package DESeq2 v. 1.26.0 [101] using read counts obtained by STAR mapping. We compared the 0, 3h and 24h time points by Wald test using the contrast option of DESeq2. Genes with |log$_2$ fold change|> 1.5 and adjusted *p*-value (padj) < 0.01 were considered differentially expressed, unless specified otherwise. Possible batch effects were inspected using Principal Component Analysis (PCA). Gene Ontology (GO) term enrichment analysis was performed using clusterProfiler package v. 3.14.3 [102]. Adjustment of *p*-values was done by Benjamini-Hochberg procedure. GO information for *C. glabrata* was obtained from CGD, while for human data we used "Genome-wide annotation for Human database" (org.Hs.eg.db) v. 3.10.0 in R. All custom calculations and visualizations were performed in R v. 3.6.1 using various packages. A differential expression data of *C. glabrata*-host interaction in the presence of albumin are provided in S2 File. To compare the gene expression levels of the host and *C. glabrata* in the medium with albumin with those in the absence of albumin, we used gene expression data from Pekmezovic *et al.* [9]. All codes, packages and their versions are available at https://github.com/Gabaldonlab/C_glabrata_with_albumin.

## Statistical analysis

Data were analysed using GraphPad Prism version 8 (GraphPad Software, La Jolla California USA). Values are presented as mean ± standard deviation (SD). If data was not distributed normally, it was Log-transformed as indicated prior to statistical analysis in GraphPad Prism. Used statistical tests are indicated in each figure legend. Statistical significance is indicated in the figures as follows: *, $p \leq 0.05$; **, $p \leq 0.01$; ***, $p \leq 0.001$; ****, $p \leq 0.0001$. All data used to generate the graphs is provided as source data (S3 File).

## Supporting information

**S1 Fig. Comparison of human and murine albumin and testing of clinical *Candida glabrata* strains. (a)** Damage of A-431 cells infected with *C. glabrata* with human or murine albumin from three different manufacturers (Sanquin Plasma Products B.V., Albuman; Bio & Sell; Sigma–for human albumin; Sigma; Innovative Research (IR); Abcam–for murine albumin). The dotted line represents the damage from A-431 cells in medium only. **(b)** Damage of A-431 cells infected with different *C. glabrata* clinical strains isolated from various anatomical sites, with or without albumin. The dotted line represents the damage from A-431 cells infected with wild type (WT) *C. glabrata* in the presence of albumin. **(c)** Dendrogram showing clustering of human (ID: Q56G89), bovine (ID: P02769.4) and murine (ID: P07724.3) albumin

protein sequences using BLASTP tool provided by NCBI. The alignment of albumin sequences is provided in S1 File, obtained using Jalview. All values are presented as mean ± SD. Damage was recorded by measuring the lactate dehydrogenase activity in the supernatant and presented as percentage of a full lysis control (A-431 treated with Triton X-100). Albumin was always used at a 5 mg/mL concentration. One-way ANOVA (a) or two-way ANOVA (b) were used to calculate statistically significant differences. Statistical significance is indicated as: $^{*}$, $p \leq 0.05$; $^{**}$, $p \leq 0.01$; $^{***}$, $p \leq 0.001$; $^{****}$, $p \leq 0.0001$. Abbreviation: BAL—bronchoalveolar lavage.
(TIF)

**S2 Fig. Expression of selected *Candida glabrata* and host genes derived from RNA-Seq data and Principal Component Analysis bi-plots. (a)** Expression of adhesin and putative adhesin *C. glabrata* genes at 3 and 24 hpi and 24 h control (24 c) in the presence of albumin, derived from RNA-Seq data (see Material and Methods). Presented values are expressed as $\log_2$ fold changes of expression compared to fungal cells in medium only at the time point 0. **(b)** Principal component analysis (PCA) bi-plot of all analysed fungal and **(c)** human samples (n = 3 for each time point; technical replicates are merged into individual samples). Control samples represent the transcriptional response of *C. glabrata* only or the host only to medium with albumin at 0 and 24 hours. Labels of the samples correspond to internal sample identifiers. **(d)** Expression of human genes involved in starvation at 3 and 24 hpi and 24 h control (24 c) in the presence of albumin, derived from RNA-Seq data (see Material and Methods). Presented values are expressed as $\log_2$ fold changes of expression compared to host cells in medium only at the time point 0.
(TIF)

**S3 Fig. Physiologically-relevant conditions in an *in vitro* infection model. (a)** Damage of A-431 infected with *C. glabrata* with or without albumin in standard conditions (RPMI pH 7–7.4, 11 mM glucose). **(b)** Damage of *Lactobacillus rhamnosus*-colonized A-431 cells infected with *C. glabrata* with or without albumin. **(c)** Damage of A-431 infected with *C. glabrata* with or without albumin at pH 4. **(d)** Damage of A-431 infected with *C. glabrata* with or without albumin in RPMI with increased glucose (20 mM) availability or without glucose (0 mM). **(e)** *C. glabrata* growth 24 h post infection of A-431 cells with or without human or murine albumin in culture medium with or without glucose. All values are presented as mean ± SD. Damage was recorded by measuring the lactate dehydrogenase activity in the supernatant and presented as percentage of a full lysis control (A-431 treated with Triton-X-100). The dotted line represents damage from uninfected A-431 cells. Albumin was always used at a 5 mg/mL concentration. One-way ANOVA was used to calculate statistically significant differences between control infection experiment **(a)** and other conditions **(b-d)**. Statistical significance is indicated as: $^{*}$, $p \leq 0.05$; $^{**}$, $p \leq 0.01$; $^{***}$, $p \leq 0.001$.
(TIF)

**S1 Table. Mutant screening data.** Damage of A-431 cells infected with wild type (WT) and different mutant strains, with or without bovine serum albumin. The values are presented as absorbance values and expressed as the percentage of damage caused by WT. Damage was recorded by measuring LDH in the supernatant.
(XLSX)

**S1 File. Alignment of protein sequences of human, bovine and murine albumin.** Comparison of human (ID: Q56G89), bovine (ID: P02769.4) and murine (ID: P07724.3) albumin protein sequences using Jalview.
(HTML)

**S2 File. Differential expression data of *Candida glabrata*-host interaction.** Differential expression data of *C. glabrata* and human cells at different time points of infection, as stated in Sheet description.
(XLSX)

**S3 File. Source data.** All data used to generate graphs in this manuscript.
(XLSX)

## Acknowledgments

We thank Dr. Oliver Bader (Institute for Medical Microbiology, University Medical Center Göttingen, Göttingen, Germany) for providing *C. glabrata* clinical isolates and Raquel Alonso-Román and Marisa Valentine (Department of Microbial Pathogenicity Mechanisms, Leibniz Institute for Natural Product Research and Infection Biology, Hans Knoell Institute, Jena, Germany) for providing lactobacilli cultures.

## Author Contributions

**Conceptualization:** Marina Pekmezovic, Selene Mogavero, Toni Gabaldón, Mark S. Gresnigt, Bernhard Hube.

**Data curation:** Marina Pekmezovic, Ann-Kristin Kaune, Sophie Austermeier, Sophia U. J. Hitzler, Hrant Hovhannisyan.

**Formal analysis:** Ann-Kristin Kaune, Sophie Austermeier, Sophia U. J. Hitzler, Hrant Hovhannisyan.

**Funding acquisition:** Mark S. Gresnigt, Bernhard Hube.

**Investigation:** Marina Pekmezovic, Mark S. Gresnigt, Bernhard Hube.

**Methodology:** Marina Pekmezovic, Ann-Kristin Kaune, Sophie Austermeier, Sophia U. J. Hitzler.

**Project administration:** Marina Pekmezovic, Toni Gabaldón, Bernhard Hube.

**Resources:** Bernhard Hube.

**Software:** Hrant Hovhannisyan, Toni Gabaldón.

**Supervision:** Selene Mogavero, Mark S. Gresnigt, Bernhard Hube.

**Validation:** Marina Pekmezovic, Selene Mogavero, Hrant Hovhannisyan.

**Visualization:** Ann-Kristin Kaune, Selene Mogavero.

**Writing – original draft:** Marina Pekmezovic, Ann-Kristin Kaune, Sophie Austermeier, Mark S. Gresnigt, Bernhard Hube.

**Writing – review & editing:** Marina Pekmezovic, Sophie Austermeier, Sophia U. J. Hitzler, Selene Mogavero, Hrant Hovhannisyan, Toni Gabaldón, Mark S. Gresnigt, Bernhard Hube.

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
