## [Decision Letter · Decision Letter 0]

17 Aug 2021

Dear Prof. Hube,

Thank you very much for submitting your manuscript "Human albumin multiplies the pathogenic potential of Candida glabrata on vaginal epithelial cells" for consideration at PLOS Pathogens. As with all papers reviewed by the journal, your manuscript was reviewed by members of the editorial board and by several independent reviewers. In light of the reviews (below this email), we would like to invite the resubmission of a significantly-revised version that takes into account the reviewers' comments.

We cannot make any decision about publication until we have seen the revised manuscript and your response to the reviewers' comments. Your revised manuscript is also likely to be sent to reviewers for further evaluation.

Sincerely,

Paul L. Fidel, Ph.D.

Guest Editor

PLOS Pathogens

Xiaorong Lin

Section Editor

PLOS Pathogens

Kasturi Haldar

Editor-in-Chief

PLOS Pathogens

orcid.org/0000-0001-5065-158X

Michael Malim

Editor-in-Chief

PLOS Pathogens

orcid.org/0000-0002-7699-2064

Comments by the Guest Editor:

Dr. Hube:

Your manuscript has been reviewed by the editorial board and two independent experts in the field. Both reviewers felt the manuscript has considerable merit. Please consider all the reviewer's comments in your rebuttal and revision. Pay particular attention to reviewer 1 who suggests some additional experimentation for confirmation of your findings. There are also additional issues to consider for comment/discussion in the revised manuscript. In addition to the review, I also suggest that you consider a different term in the title for 'multiplies'. This term implies a quantitative value of sorts. A more qualitative term would be more appropriate. Second, due to the potential role of glucose in the human albumin-associated effects on vaginal epithelial cells, you should consider including a few comments on this effect relative to diabetes. Women with diabetes tend to have higher prevalence of VVC or RVVC caused by C. glabrata. This could be in part due to the higher presence of glucose available.

Reviewer's Responses to Questions

**Part I - Summary**

Reviewer #1: Pekmezovic et al. address the effect of human serum albumin on the interaction of C. glabrata with vaginal epithelial cells. They find that human and bovine – but not mouse – albumin promote proliferation of C. glabrata in the presence of vaginal epithelial cells, as well as damage to the epithelial cells by C. glabrata. Albumin increases adherence of C. glabrata to epithelial cells. The cytotoxicity requires direct contact to the epithelial cells, but the increased C. glabrata proliferation does not: it also occurs in a transwell setup. Protease inhibitors, and inhibition of albumin uptake by epithelial cells, abolish the increased cell damage, suggesting that albumin uptake and processing by the cells mediates increased sensitivity to C. glabrata. Protease inhibitors also abolish the increased C. glabrata proliferation.

Expression profiling in C. glabrata indicated an increase in oxidative stress resistance genes, and indeed C. glabrata cells grown in the presence of epithelial cells with albumin are more resistant to H2O2. A screen for C. glabrata mutants that cause less epithelial cell damage in spite of having no reduced growth capacity identified HAP5, encoding an iron-regulated transcription factor. Transferrin and BPS abolish the albumin-stimulated growth and the damage potential of C. glabrata, leading the authors to the suggest that iron is responsible for the albumin-mediated stimulation of C. glabrata proliferation and damage to epithelial cells.

Evaluation: the question of how to best model virulence of pathogens in vitro is key in the field of microbial pathogenesis. Therefore, the identification of human albumin as a specific host factor involved in virulence of a fungal pathogen is an important finding. This is a well-written paper in which the authors show through a series of convincing experiments that the mechanism by which albumin increases C. glabrata-induced damage to epithelial cells involves epithelial cell uptake and processing of albumin. A possible involvement of iron release or uptake, while being an attractive hypothesis based on the phenotype of the hap5 mutant, is less convincing at this point (see comments below).

Reviewer #2: The manuscript by Pekmezovicet al describes a careful work on the role of human (and bovine) albumin on thepathogenesis of the important human fungal pathogen, Candida glabrata. It iswell known, that C. glabrata causes quite frequently candidiasis in humans(actually it is the second or third most common cause of this disease), surprisinglyhowever this species is often almost avirulent in different in vivo and in vitromodel systems. The authors investigated the biological background of thiscontradiction and found that human albumin (digested by the human epithelialcells) can increase the growth, adhesion, and damage-causing capacity of C.glabrata cells. The screening of a deletion library revealed that Hap5, animportant regulator of the iron homeostasis, is a key regulator of the albumin-dependentvirulence enhancement. The manuscript is very well written and easy to follow.

**Part II – Major Issues: Key Experiments Required for Acceptance**

Reviewer #1: 1) Role of iron: in Fig. 6b, 6c, the authors interpret the abolition of albumin-stimulated cytotoxicity and C. glabrata growth by iron chelators as evidence that this stimulation involves iron supply. However, an alternative explanation is that albumin relaxes another, unknown limiting factor for proliferation and epithelial cell damage, whereas addition of chelators impose iron withholding as a new limiting factor.

Furthermore, the induction seen in Fig. 6d of an iron starvation response in C. glabrata exposed to epithelial cells with albumin (as evidenced by upregulation of iron uptake genes such as FTR1, SIT1, FTH1) suggests that iron is withheld, rather than supplied, under these conditions, contradicting the notion that improved growth can be attributed to an increased iron supply.

Thus, the model that iron supply is driving the increased C. glabrata proliferation and cytotoxicity in the presence of albumin is not well supported by the data. The notion that iron supply is a limiting factor to begin with could be corroborated by testing the effect of direct iron addition to the RPMI medium.

2) Induction of Hap5-dependent iron-regulated genes by albumin: to my understanding, the experiment shown in Fig 6d doesn’t address this (i.e., there is no comparison +/- albumin). In order to make the point that it is albumin addition that induces the transcriptional response, incubation of C. glabrata in medium with epithelial cells, with vs. without albumin, should be compared, rather than comparing medium + epithelial cells + albumin vs. medium alone. I.e., the effects of albumin should be tested by comparing conditions +/– albumin, everything else remaining equal.

Likewise for Fig. 7: the gene expression in epithelial cells seems to compare only +/- C. glabrata in the presence of albumin, not infection with C. glabrata +/- albumin. An analysis of the effect of albumin alone on the cells is shown after 24h, however the comparison is to t0. Comparison to an identical 24h culture without albumin would have been more adequate.

Reviewer #2: (No Response)

**Part III – Minor Issues: Editorial and Data Presentation Modifications**

Reviewer #1: ll. 115-116: what’s the MOI in albumin-free medium?

ll. 300-302: “Albumin can bind iron and was also demonstrated to deliver iron to C. albicans via the CFEM hemophore relay network (21,33)”: this should be qualified, because albumin was shown to bind and deliver heme, not free iron. There is no indication of specific iron binding by serum albumin to my knowledge (the iron binding protein in serum is transferrin, as mentioned later). Furthermore, there is no evidence for the existence of a CFEM hemophore system in C. glabrata.

Fig. 1c: the similarity tree of the three albumins isn’t very informative. However the availability of a non-stimulating albumin, MSA, might enable the authors to test their suggestion that HSA and BSA function by binding and supplying iron: iron binding could be directly tested (e.g., retention of radioactive iron by membrane-bound albumin), with MSA providing a negative control.

Reviewer #2: I only have some minor suggestions and questions:

1. The normal vaginal pH is between 3.8 and 4.5, the authors used RPMI for the cytotoxicity assays, which has a higher pH level (7.2-8.3 depending of the manufacturer) did the authors considered this factor, since the pH level of the environment can influence the virulence related properties?

2. The vagina has a complex and well-studied microbial community, predominantly composed of Lactobacillus species. Did the authors considered this microbial components of the vagania as a potential factor that can influence C. glabrata viruelnce and pathogenesis in the presence of the human albumin? I suggest at least mention this important aspect in the discussion.

3. The increased expression of the adhesion related genes at the early timepoint of the albumin exposeur is an interesting finding, did the authors further tested the adhesion ability of the C. glabrata cells in the presence of human albumin? The increased expression of genes of adhesins and putative adhesins caused indeed an increased attachemnt to the host cells?

4. During the screen of the mutant library, some deletion mutants showed an increased damage capacity in compare to the control, is there any speculation about the reason of this results?

PLOS authors have the option to publish the peer review history of their article (what does this mean?). If published, this will include your full peer review and any attached files.

Reviewer #1: No

Reviewer #2: No
---

## [Editor Report · Decision Letter 1]

15 Oct 2021

Dear Professor Hube,

We are pleased to inform you that your manuscript 'Human albumin enhances the pathogenic potential of Candida glabrata on vaginal epithelial cells' has been provisionally accepted for publication in PLOS Pathogens.

Please also note two small edits (detailed below) that I am suggesting for the Discussion as clarification, and to be added during your review of the proof.

Best regards,

Paul L. Fidel, Ph.D.

Guest Editor

PLOS Pathogens

Xiaorong Lin

Section Editor

PLOS Pathogens

Kasturi Haldar

Editor-in-Chief

PLOS Pathogens

orcid.org/0000-0001-5065-158X

Michael Malim

Editor-in-Chief

PLOS Pathogens

orcid.org/0000-0002-7699-2064

Professor Hube:

You have adequately responded to the critique and revised the manuscript accordingly. The manuscript is much improved. While the editor will provisionally accept the manuscript, as the Guest Editor, I have two small edits I suggest in the Discussion to help clarify the point being made regarding the glucose results. In line 496 I suggest you end the sentence .....and no tissue damage in the presence of albumin. And in line 497 I suggest you describe the C. glabrata pathogenicity as....role in 'albumin-enhanced' C. glabrata pathogenicity. If you agree we are asking that you add these edits when you receive the proof. They likely won't be in the queries for the proof but we ask that make a note to add these as minor edits when you do receive the proof.
---

## [Editor Report · Acceptance letter]

24 Oct 2021

Dear Prof. Hube,

We are delighted to inform you that your manuscript, "Human albumin enhances the pathogenic potential of *Candida glabrata* on vaginal epithelial cells," has been formally accepted for publication in PLOS Pathogens.

Best regards,

Kasturi Haldar

Editor-in-Chief

PLOS Pathogens

orcid.org/0000-0001-5065-158X

Michael Malim

Editor-in-Chief

PLOS Pathogens

orcid.org/0000-0002-7699-2064